# Inhibitory Effect and Mechanism of *Trichoderma taxi* and Its Metabolite on *Trichophyton mentagrophyte*

**DOI:** 10.3390/jof8101006

**Published:** 2022-09-25

**Authors:** Chenwen Xiao, Lin Li, Yan Liu, Yee Huang, Yanli Wang, Jiaoyu Wang, Guolian Bao, Guochang Sun, Fucheng Lin

**Affiliations:** 1State Key Laboratory for Managing Biotic and Chemical Treats to the Quality and Safety of Agro-Products, Institute of Animal Husbandry and Veterinary Science, Zhejiang Academy of Agricultural Sciences, Hangzhou 310021, China; 2State Key Laboratory for Managing Biotic and Chemical Treats to the Quality and Safety of Agro-Products, Institute of Plant Protection and Microbiology, Zhejiang Academy of Agricultural Sciences, Hangzhou 310021, China

**Keywords:** *Trichophyton mentagrophytes*, trichodermin, *Trichoderma taxi*, endophytic fungi, ABC gene

## Abstract

*Trichophyton mentagrophytes* is an important zoonotic dermatophyte, which seriously harms the skin of humans and animals. Chemical drugs are generally used for the prevention and treatment of the disease caused by *T*. *mentagrophytes*. Discovering new compounds from natural products is an important approach for new drug development. *Trichoderma* includes a variety of fungal species used for biological control of phytopathogenic fungi. However, the antifungal effects of *Trichoderma* and their metabolites on zoonotic fungal pathogens are largely unknown. Here, the effect of trichodermin, a metabolite derived from the plant endophytic fungus *Trichoderma taxi*, on *T. mentagrophytes* was examined, and the underlying mechanism was explored. *T. mentagrophytes* growth was suppressed significantly by trichodermin and completely inhibited under 1000 μg/mL trichodermin. The production and germination of *T. mentagrophytes* spores were remarkably reduced upon exposure to trichodermin, in comparison with control samples. Treatment of lesions caused by *T. mentagrophytes* on the rabbit skin with 1 mg/mL trichodermin prompted the healing process significantly; however, 20 mg/mL trichodermin was likely toxic to the skin. Under trichodermin treatment, the number of mitochondria in *T. mentagrophytes* increased significantly, while a few mitochondria-related genes decreased, indicating possible mitochondrial damage. In transcriptome analysis, the GO terms enriched by DEGs in the trichodermin-treated group included carbohydrate metabolic process, integral component of membrane, intrinsic component of membrane, and carbohydrate binding, while the enriched KEGG pathways comprised biosynthesis of secondary metabolites, glycolysis/gluconeogenesis, and carbon metabolism. By comparing the wild type and a gene deletion strain of *T. mentagrophytes*, we found that *CDR1*, an ABC transporter encoding gene, was involved in *T. mentagrophytes* sensitivity to trichodermin.

## 1. Background

*Trichophyton mentagrophytes* is an important zoonotic skin pathogen, which infects the skin corneum and associated keratinization tissues, causing harm to both humans and animals [1]. *T. mentagrophytes*-associated skin diseases are mainly treated with chemical drugs [2]. Although there are multiple clinical drugs available [3], their efficacy is instable, and the disease is likely to relapse due to drug tolerance arising from long-term and repeated treatment. Hence, it is urgent to develop novel, efficient, and long-acting drugs.

Identifying novel compounds from natural products is an important approach for developing new drugs. Microorganisms are considered a major source of natural products. For example, a variety of fungi in the *Trichoderma* genus can be used for biological control of plant diseases. The biological control mechanisms in *Trichoderma* include hyperparasitism, antibiosis, and competition. The antibiotic substances produced by these organisms are critical in biological control. The separation, identification and application of endophytic fungi [4] is an emerging research area [5,6]. Previous findings revealed that the endophytic fungus *Trichoderma taxi* strain ZJUF0986 and its substance trichodermin are effective in inhibiting multiple plant pathogenic fungi [7].

In this study, trichodermin, a metabolite from *T. taxi* strain ZJUF0986, was examined for its antifungal effect and mechanism on *T. mentagrophytes*. The effects of trichodermin on *T. mentagrophytes* were evaluated by examining culture growth, spore germination, and electron microscopy analysis. The molecular mechanism of trichodermin inhibition on *T. mentagrophytes* was investigated by transcriptome sequencing and qRT-PCR. Animal skin inoculation was performed to test the curative potential of trichodermin on skin disease caused by *T. mentagrophytes*, and digital quantitative PCR was used to assess fungal development in skin tissue. In addition, fluorescent protein-labeled strains were established and utilized to monitor the effect of *T. taxi* on *T. mentagrophytes* in culture medium. These findings provide evidence for using traditional biological agents for plant pathogens in the control of animal skin disease caused by *T. mentagrophytes*. 

## 2. Methods and Materials

### 2.1. Drug Sensitivity for Colony Size and Spore Generation

The preparation process of trichodermin was as follows. After the strain ZJUF0986 was activated on the PDA medium, the fungal mycelium at the edge of the colony was cut with a whole puncher and added to the liquid medium for culture and fermentation.

The crude petroleum ether extract of the fermentation broth was dissolved in petroleum ether, and back extracted with deionized water. Anhydrous sodium sulfate was added to the organic phase and dried. After filtration, the extract was concentrated under reduced pressure at 50 °C to yield a colorless viscous substance. The colorless viscous substance was then dissolved with chloroform, filtered, mixed with H60 thin layer chromatography silica gel. After the solvent was removed, column chromatography was performed with chloroform/methanol as mobile phase with different mixture compositions. The collected eluates for each unit were concentrated and dissolved in methanol. The spots were tracked by high-performance thin layer chromatography (TLC). Samples with the same spots and antibacterial activity were combined to separate and purify active metabolites [7].

SDA plates were prepared with trichoderm in at 10, 100 and 1000 μg/mL. *T. mentagrophytes* was inoculated on these plates and cultured for 6 days. Then, the colony sizes were measured. The colonies were washed with normal saline and filtered with two layers of sterilized microscope lens paper to harvest spores, which were counted on a blood cell count board. The experiments were performed three times with three duplicates in each.

### 2.2. Spore Germination of T. mentagrophytes under Trichodermin

Totally 5 × 10^5^/mL *T. mentagrophytes* spores were added to solutions containing 0, 10, and 100 μg/mL trichodermin, respectively, and cultured at 28 °C in the dark for 14 h. Spore germination was assessed by Calcofluor white staining as follows. The spore solution was dripped onto a glass plate, stained with 10 mg/mL Calcofluor white solution, and observed immediately by fluorescence microscopy. 

### 2.3. Microscopic Examination of T. mentagrophytes under Trichodermin

SDA plates containing 10 and 100 μg/mL trichodermin were prepared, respectively. *T. mentagrophytes* was cultured on these plates for 6 days before ultrastructural analysis by transmission electron microscopy. Meanwhile, *T. mentagrophytes* underwent mitochondrial staining with the MitoTracker Green^FM^ green dye (lot no. M7514, Invitrogen, Carlsbad, CA, USA) and were observed by a Zeiss LSM880 confocal microscope (Zeiss, Germany). MitoTracker Green^FM^ green is a bright green fluorescence probe (F488 nm) that hardly fluoresces in aqueous solutions and only fluoresces when it accumulates in the mitochondrial lipid environment. Before use, 1 mM stock solution of the dye was prepared with DMSO according to the kit’s instructions and further diluted with water (1000×) to prepare the staining solution. The mycelia were placed onto a glass slide, stained with 15 μL of staining solution, and observed immediately under a confocal microscope (Zeiss, Germany).

### 2.4. Interaction between Trichoderma Taxi and T. mentagrophytes

*T. taxi* strain ZJUF0986 and *T. mentagrophytes* were co-inoculated on an SDA plate by keeping an interval of 5 cm. After 3 days, the colonies were close to each other. The interaction between them was assessed by light microscopy on an Axio Imager A2 (Zeiss). 

### 2.5. Transcriptome Sequencing

SDA plates with trichodermin at 0, 10, and 100 μg/mL were prepared, respectively. The fungal colonies were prepared by the method of fungal colony drilling and cultured on the plates for 6 days. Then, the mycelia were collected and stored in liquid nitrogen for high-throughput sequencing. RNA Nano 6000 Assay Kit was utilized for integrity assessment of RNA samples on a Bioanalyzer 2100 system (Agilent Technologies, Santa Clara, California, USA).

In total, 1 μg RNA/sample was utilized as input material, and library preparation was carried out as described in a previous study [8]. In brief, mRNA isolation from total RNA employed poly-T oligo-linked magnetic beads. After fragmentation with divalent cations at high temperature, first- (random hexamer primers and M-MulV Reverse Transcriptase (RNase H-)) and second-strand (DNA Polymerase I and RNase H) cDNAs were synthesized. The exonuclease and polymerase activities of DNA Polymerase I converted the remaining overhangs into blunt ends. Then, the 3′ termini of DNAs were adenylated, and an adaptor was ligated for hybridization. For selecting cDNAs of 370~420 bp, the AMPure XP system (Beckman Coulter, Beverly, USA) was utilized. Next, PCR was carried out using Phusion High-Fidelity DNA polymerase, Universal PCR primers, and index (X) primer, followed by PCR product purification (AMPure XP system) and library quality assessment on an Agilent Bioanalyzer 2100 system. TruSeq PE Cluster Kit v3-cBot-HS (Illumina) was utilized to cluster index-coded specimens on a cBot Cluster Generation System as directed by the manufacturer. Then, the prepared libraries underwent sequencing on an Illumina Novaseq platform, with 150-bp paired-end reads generated.

Raw reads in the fastq format first underwent processing using in-house perl scripts, generating clean reads without adapter or ploy-N of high quality. Meanwhile, Q20, Q30, and GC contents of clean reads were determined. Only these high-quality reads were further examined. Reference genome and gene model annotation files were retrieved from the genome website. The reference genome’s index was obtained with Hisat2 version 2.0.5, and alignment of paired-end reads to the reference genome utilized Hisat2 version 2.0.5. Feature Counts version 1.5.0-p3 was utilized for counting reads mapped to various genes. Next, the FPKM (fragments per kilobase of transcript sequence per million base pairs) index was derived according to gene length and the number of reads mapped to the indicated gene. FPKM is considered the most accurate index for assessing gene expression after sequencing.

Differential expression analysis between 2 duplicate treatment groups used the DESeq2 R package (v1.20.0). The Benjamini and Hochberg’s method was utilized to control the false discovery rate. Adjusted *p* < 0.05 indicated a differentially expressed gene (DEG). Gene Ontology (GO) analysis of DEGs employed cluster Profiler in R, with corrected gene length bias. GO terms showing corrected *p* < 0.05 were deemed to be significantly enriched by DEGs. KEGG (http://www.genome.jp/kegg/) (accessed on 11 June 2021) analysis was performed with cluster Profiler in R. The Duncan’s test was carried out for data analysis with a minimum of n = 3. *p* < 0.05 indicated statistical significance.

### 2.6. Real-Time PCR Analysis

Totally 1μg of total RNA was utilized for reverse transcription with a first strand cDNA synthesis kit manufactured by Promega (Madison, Wisconsin, USA) as directed by the manufacturer. An ABI StepOnePlus (Applied Biosystems, Waltham, Massachusetts, United States) was employed for qRT-PCR with SYBR Green Supermix (TaKaRa, Kusatsu, Shiga, Japan) as proposed by the manufacturer. Amplification was performed at 94 °C (10 min), with subsequent 40 cycles at 94 °C (15 s) and 60 °C (31 s), and final extension. The 2^−ΔΔCt^ method [9] was utilized to analyze triplicate assays, with 18S as a reference gene. Relative mRNA levels were presented as mean ± SD. We selected 8 candidate genes with statistically significant expression differences in the transcriptomic data, designed primers according to their gene sequences and verified their expression changes by fluorescence quantitative PCR. The primers applied in qRT-PCR are listed in Table 1.

### 2.7. Interaction of T. taxi and T. Mentagrophytes

*T. taxi* strainZJUF0986 and *T. mentagrophytes* were labeled with GFP and RFP, respectively [10]. The pHMG and pHMR1 plasmids expressing GFP and DsRED-PTS1 were introduced into *T. taxi* and *T. mentagrophytes*, respectively. Then GFP-labeled *T. taxi* and RFP-labeled *T. mentagrophytes* were co-cultured on SDA plates for 6 days. The interaction of the two strains was observed under a confocal microscope (Zeiss, LSM880, Germany).

### 2.8. Assessment of Trichodermin’s Effect on Animal skin Disease

Male New Zealand rabbits (5 week sold) weighing 1–1.5 kg were purchased from Zhejiang Animal Center, Zhejiang Academy of Agricultural Sciences (Hangzhou, China), and acclimatized for 1 week prior to the study.

The rabbit skin was first inoculated with *T. mentagrophytes*. Then, diseased individuals were used for therapeutic testing. Treatment was conducted by smearing 1000 µg/mL and 20 mg/mL trichodermin once a day (2 mL each) for 3 consecutive days. The treatment effect was observed by photography after 14 days of treatment. 

This study was approved by the Ethics Committee of the Zhejiang Academy of Agricultural Sciences (ethics protocol no. 002762) and performed in accordance with the principles and guidelines of the Zhejiang Farm Animal Welfare Council of China. 

### 2.9. Involvement of the CDR1 Gene in the Sensitivity of T. mentagrophytes to Trichodermin

*CDR1* deletion strains were generated as follows. The 1.2 kb *CDR1* upstream and 1.1 kb downstream fragments were amplified, respectively, by PCR, with the genomic DNA of *T. mentagrophytes* as a template. The two fragments were inserted, respectively, into p1300-KO to generate*CDR1* knockout vector pKO-TmCDR1, which was introduced into *T. mentagrophytes* via *At*MT, and the transformants were screened on CM plates containing 250 μg/mL hygromycin B. The *CDR1* deleted strains T9-21 and T4-12 were confirmed by multiple-genome PCR and real-time quantitative PCR. The TmCDR1 gene was re-introduced into the T9-21 strain to obtain the complementary strains PNMNR6 (5) and PNMG7 (6). 

The *T. mentagrophytes* wild-type strains T9-21 and T4-12 and the CDR1-complemented strains PNMNR6 (5) and PNMG7 (6) were inoculated on SDA plates containing 10 and 100 μg/mL trichodermin for a 6-day culture. The experiments were repeated 6 times. 

### 2.10. Digital PCR for Assessing the Amount of Fungal Material in Skin Tissue

On the 14th day of the animal skin inoculation experiment, skin samples from the lesions, treated with 1000 µg/mL trichodermin or untreated controls, were collected. An automatic digital PCR all-in-one machine (Sniper, DQ24) was used to detect the amounts of fungal material in skin tissue. Primer sequences and reagent information are provided in Table 2 and Table 3. PCR was performed at 60 °C (5 min) and 94 °C (15 min), followed by 40 cycles of 94 °C (20 s) and 63 °C (30 s) and extension for 30 s. Triplicate assays were performed.

After the end of the experiment, all animals were euthanized by intravenous injection of 100 mg/kg of sodium pentobarbital (Sigma-Aldrich, St.Louis, MO, USA, CASno: 57-33-0), resulting in the painless death of experimental animals.

## 3. Results

### 3.1. Colony Growth and Spore Generation in T. mentagrophytes Are Affected by Trichodermin

*T. mentagrophytes* colonies were gown under trichodermin, and differences in *T. mentagrophytes* colony sizes and spore numbers were evaluated after 6 days of culture. Trichodermin at 1000 μg/mL completely inhibited the growth of *T. mentagrophytes* (Figure 1A). With decreasing concentration of trichodermin, *T. mentagrophytes* colonies increased slowly but remained significantly smaller compared with the SDA (blank control) group. The spores generated by the colonies under 100 μg/mL and 10 μg/mL trichodermin were remarkedly reduced in comparison with the control group (*p* < 0.01) (Figure 1B,C).

### 3.2. Spore Germination of T. mentagrophytes under Trichodermin

Spore germination *T. mentagrophytes* were examined after a 14 h incubation with various amounts of trichodermin by fluorescence microscopy in combination with Calcofluor white staining (Figure 2A). There was a concentration-dependent decrease in spore germination rate by trichodermin (Figure 2B). Spore germination rates in the 10 μg/mL and 100 μg/mL trichodermin groups were 13.3% and 3.9%, respectively (Table 4), which were significantly lower than the control value (43.5%); spore germination inhibition rates for 10 μg/mL and 100 μg/mL trichodermin were 69% and 91%, respectively (Table 4), and the degrees of spore germination inhibition were correlated with the concentration of trichodermin. ** stand for *p* < 0.01; * stand for *p* < 0.05.

### 3.3. Electron and Confocal Microscopy Features of T. mentagrophytes under Trichodermin

The ultrastructural properties of *T. mentagrophytes* after incubation with 10 and 100 μg/mL trichodermin were analyzed by TEM. After treatment with trichodermin, the mitochondria of *T. mentagrophytes* were swollen (Figure 3A). After staining with the mitochondrial dye MitoTracker Green^FM^, mitochondria in *T. mentagrophytes* mycelia were counted (Figure 3B). The mitochondria of *T. mentagrophytes* treated with trichodermin at 100 μg/mL increased significantly compared with the control group.

### 3.4. Interaction of T. taxi on T. mentagrophytes

*T. taxi* and *T. mentagrophytes* were co-inoculated on SDA plates by keeping an interval of 5 cm in between. After 3 days of culture, the colonies were closer to each other. The interaction between them was observed microscopically. The hyphae of *T. taxi* were detected that extended to and intertwined with the mycelia of *T. mentagrophytes*, and some hyphae of *T. taxi* invaded those of *T. mentagrophytes* (Figure 4A). With fluorescent labeling, the interaction between *T. taxi* (GFP-labeled) and *T. mentagrophytes* (RFP-PTS1-labeled) was observed by confocal microscopy (Figure 4B). GFP and RFP signals could better depict the interaction and mycelial twisting between *T. taxi* and *T. mentagrophytes*.

### 3.5. Transcriptome Analysis

To further reveal the mechanisms of the effect of trichodermin on *T. mentagrophytes*, transcriptome analysis was performed. The transcriptomic data were submitted to GEO (GEO accession numbers: BioProject, PRJNA785595; SRA, SRP348930). After treatment with trichodermin at 100 μg/mL, significant differentially expressed genes (DEGs) compared with the control group were determined at *p* < 0.05 (Figure 5A,B). Totally1681 DEGs were found in the 100 μg/mL trichoderm group compared with the control group (Figure 5A), including 897 up-regulated and 784 down-regulated genes (*p* < 0.05). Meanwhile, 813 DEGs were found after treatment with trichoderm at 100 μg/mL compared with the 10 μg/mL trichodermin group, including 405 up-regulated and 408 down-regulated genes (*p* < 0.05). In total, 196 DEGs were found between the 10 μg/mL trichodermin and control groups, including 86 up-regulated and 110 down-regulated genes (*p* < 0.05). The heatmap of the DEGs showed that the gene expression profile of the 10 μg/mL trichodermin group was similar to that of the control group, while the profile of the 100 μg/mL trichodermin group was obviously different from those of the control and 10 μg/mL trichodermin groups. 

Based on GO analysis, treatment with trichodermin at 100 μg/mL resulted in enrichment in GO terms such as carbohydrate metabolic process, integral component of membrane, intrinsic component of membrane, and carbohydrate binding and transporter activity (Figure 6). The most enriched GO terms after treatment with 10 μg/mL trichodermin were carbohydrate metabolic process, integral component of membrane, intrinsic component of membrane, and carbohydrate binding. The most enriched KEGG pathways in the 100 μg/mL and10 μg/mL trichodermin treatment groups were biosynthesis of secondary metabolites, glycolysis/gluconeogenesis, carbon metabolism, pentose phosphate pathway, and fatty acid metabolism.

### 3.6. RT-PCR Verification of Transcriptomic Data

In qRT-PCR, gene expression changes corroborated transcriptomic data, indicating transcriptomic data were reliable and could be used for further screening (Figure 7A). This analysis demonstrated mitochondria-related genes were significantly up-regulated after treatment with trichodermin at 100 μg/mL, while ATP-related genes were down-regulated; the ABC transporter gene TMEN-3870 was down-regulated under trichodermin exposure at 100 μg/mL (*p* < 0.05, Figure 7B).

### 3.7. Impact of the CDR1 Gene on Fungal Sensitivity to Trichodermin 

After 6 days of culture, the *CDR1* deletion strains T9-21 and T4-12 were significantly inhibited by 10 and 100 μg/mL trichodermin compared with wild-type strains, while this hypersensitivity was recovered in the *CDR1* complemented strains PNMNR6 (5) and PNMG7 (6) (Figure 8). 

### 3.8. Trichodermin Alleviates PM-Associated Skin Disease in Rabbits

After treatment by smearing 1000 µg/mL trichodermin, *T. mentagrophytes*-associated skin disease showed significantly improved wound healing. However, the toxic injury to the skin was induced by trichodermin at 20 mg/mL. Digital PCR showed that *T. mentagrophytes* organisms in the treatment group were basically eliminated after treatment with *Trichoderma*. The therapeutic effect was ideal (Figure 9).

## 4. Discussion

Based on U.S. Environmental Protection Agency (EPA) data, about 244,000 and 37,000 tons of fungicides were utilized globally and in the USA in 1997, respectively [11]. Azoles show inhibitory activities in diverse fungi and are widely used to prevent and treat fungal diseases. As observed with antibiotics, fungal resistance to antifungal drugs might result from the increasing application of these drugs in humans, animals, and plants [12,13]. Therefore, developing novel antifungal drugs is highly required.

*Trichoderma* covers diverse fungal species widely found in rotten wood, litter, soil, organic fertilizers, plants, and air. As a biological control resource, quite a few *Trichoderma* species have been used in the control of plant fungal pathogens in recent decades [14,15]. Its biological control mechanism includes competition, hyperparasitism, antibiosis, plant resistance induction, etc. [16]. In addition, *Trichoderma* can also change the structure of the soil microbial community and improve the contents of effective nutrients in the soil. Extensive research on *Trichoderma* is inseparable from the broad-spectrum and efficient antibacterial activity of its main metabolite, trichodermin. However, due to cytotoxicity [17,18], its use for biological control is restricted. The inhibitory effects of *Trichoderma* and its metabolite on animal-derived fungi have not been reported. In this study, trichodermin extracted from a plant endophytic fungus, *T. taxi* [19], was used to treat rabbit skin mycosis caused by *T. mentagrophytes*. At the same time, the antifungal mechanism of trichodermin and *T. taxi* against *T. mentagrophytes* was examined.

From transcriptomic data, the most common enriched GO terms in the 100 μg/mL and 10 μg/mL trichodermin treatment groups were carbohydrate metabolic process, integral component of membrane, intrinsic component of membrane, and carbohydrate binding. The most common enriched KEGG pathways in the 100 μg/mL and 10 μg/mL trichodermin treatment groups were biosynthesis of secondary metabolites, glycolysis/gluconeogenesis, and carbon metabolism. These results showed that trichodermin significantly affects the composition and metabolism of the cell membrane when it alters the spores. In addition, it could also affect the adhesion of some carbohydrates. Carbohydrates have different biological roles: they provide energy to cells, are responsible for the activities of some proteins, and may act as recognition sites. The reaction of trichodermin may be related to changes in cell polarity to influence the adhesion of some carbohydrates. Upon phagocytosis by macrophages, *C. albicans* induces carbon metabolism pathways, including gluconeogenesis, the glyoxylate cycle, and fatty acid β-oxidation. Therefore, the effects of alternative carbon sources on stress resistance and virulence in *C. albicans* are highly associated [20]. In particular, glycosyl phosphat idylinositol (GPI)-anchored proteins are known to respond to pH in the environment [21]. The present study suggested carbon metabolism as one of the most important responses of *T. mentagrophytes* to trichodermin stimulation since the fungi should face changes in pH and growth conditions.

As for in vitro antifungal data, *T. mentagrophytes* colonies were completely inhibited by trichodermin at 1000 μg/mL. This finding demonstrates that trichodermin at 1000 μg/mL could completely inhibit the growth of *T. mentagrophytes*. After treatment with 100 μg/mL and 10 μg/mL trichodermin, the numbers of *T. mentagrophytes* spores decreased significantly with a dose-dependent relationship, corroborating previous findings [22]. The active metabolites of ZJUF0986 in a previous report significantly affected spore germination in *Magnaporthe oryzae* and delayed the formation of infection thrombus of *M. oryzae*. Under stimulation by trichodermin, mycelial deformation of *T. mentagrophytes* was observed by electron microscopy and confocal microscopy. An interesting phenomenon is that under the electron microscope, the mitochondria in the mycelium were expanded. The mitochondria stained with a green dye were scattered, which may indicate mitochondrial degradation. Studies have reported the antifungal mechanism of *Trichoderma viride* metabolites against *Aspergillus niger* and litchi anthrax. The vacuoles in *Litchi anthrax* hyphae were expanded and deformed, with the mitochondria and chloroplasts in Litchi anthracnose hyphae treated with metabolites differentiated into fine particles [23], which is consistent with the finding of this study. Combined with the above transcriptomic data, we found mitochondrial genes were significantly down-regulated by trichodermin. This finding suggests that mitochondria are obviously damaged during trichodermin treatment. It was reported that bacterial infection increases MDA and ROS contents in *Musca domestica*, decreases the mitochondrial membrane potential, decreases the copy number of mtDNA, and increases 8-oxoG content, all indicating significant mitochondrial damage. Bacterial stimulation causes changes in complex activity and oxidative phosphorylation efficiency in *Musca domestica* mitochondrial respiratory chain; the organisms synthesize enough ATP by increasing oxidative phosphorylation efficiency to resist the invasive effect caused by bacterial stimulation [24]. In this study, based on electron microscopy, confocal microscopy, and transcriptomic data, we speculate that trichodermin can cause mitochondrial damage and reduce the efficiency of the respiratory chain, but *T. mentagrophytes* can synthesize enough ATP by increasing the related ATP synthase to resist the invasive effect of trichodermin stimulation. In addition, we also found that some ABC genes in *T. mentagrophytes* were down-regulated by trichodermin. Therefore, we considered knocking out one of the ABC genes, *CDR1*, to further examine the effect of trichodermin on *T. mentagrophytes* in the absence of this gene. The results were exciting, with the drug sensitivity test showing that the *CDR1* gene plays a key regulatory role in the effect of trichodermin on *T. mentagrophytes*.

In the subsequent animal skin infection experiment, trichodermin could treat rabbit skin mycosis caused by *T. mentagrophytes*, although a high dosage should be avoided: it could cause rabbit skin cracking at 20 mg/mL, indicating that it could induce skin irritation. Documents have reported that trichodermin metabolites have phytotoxic effects [25]. The extracellular metabolites and conidia of *Trichoderma koningii* smf2 were tested by acute oral toxicity test in a mouse model, as well as skin and eye irritation tests in rabbit models. It was found that the extracellular metabolites and conidia of *T. koningii* smf2 were practically non-toxic to mice and showed no skin irritation in rabbits [26]. The latter results corroborated this study. At an appropriate dose, trichodermin has no obvious irritative effect on the skin. However, at high doses, skin irritation is relatively pronounced.

Gao et al. investigated the hyperparasitic activities of several *Trichoderma* species in *Rhizoctonia solani*, *Valsa sordida*, *Botryosphaeria ribis, Valsa ceratosperma,* and *Botryosphaeria berengeriana* [27]. Antagonism culture assays revealed that in most cases, *Trichoderma* came into contact with the pathogen two days after inoculation and then covered or invaded the microbial colony to inhibit its growth. Light microscopy and scanning electron microscopy showed that *Trichoderma* had different ways of hyperparasitism in different plant pathogenic fungi. *Trichoderma* could wrap around, grow parallel to, or intertwine along the mycelium of the pathogen. In this study, we also found that *T. taxi* and *T. mentagrophytes* were intertwined and involved with each other.

In summary, a plant endocytic fungus, *T. taxi*, and its metabolite trichodermin inhibit the animal dermatosis fungus *T. mentagrophytes*, and could be considered biological agents for the skin disease caused by this fungus. This is one of the few studies assessing the effects of plant biocontrol fungi on animal pathogens.

## Figures and Tables

**Figure 1 jof-08-01006-f001:**
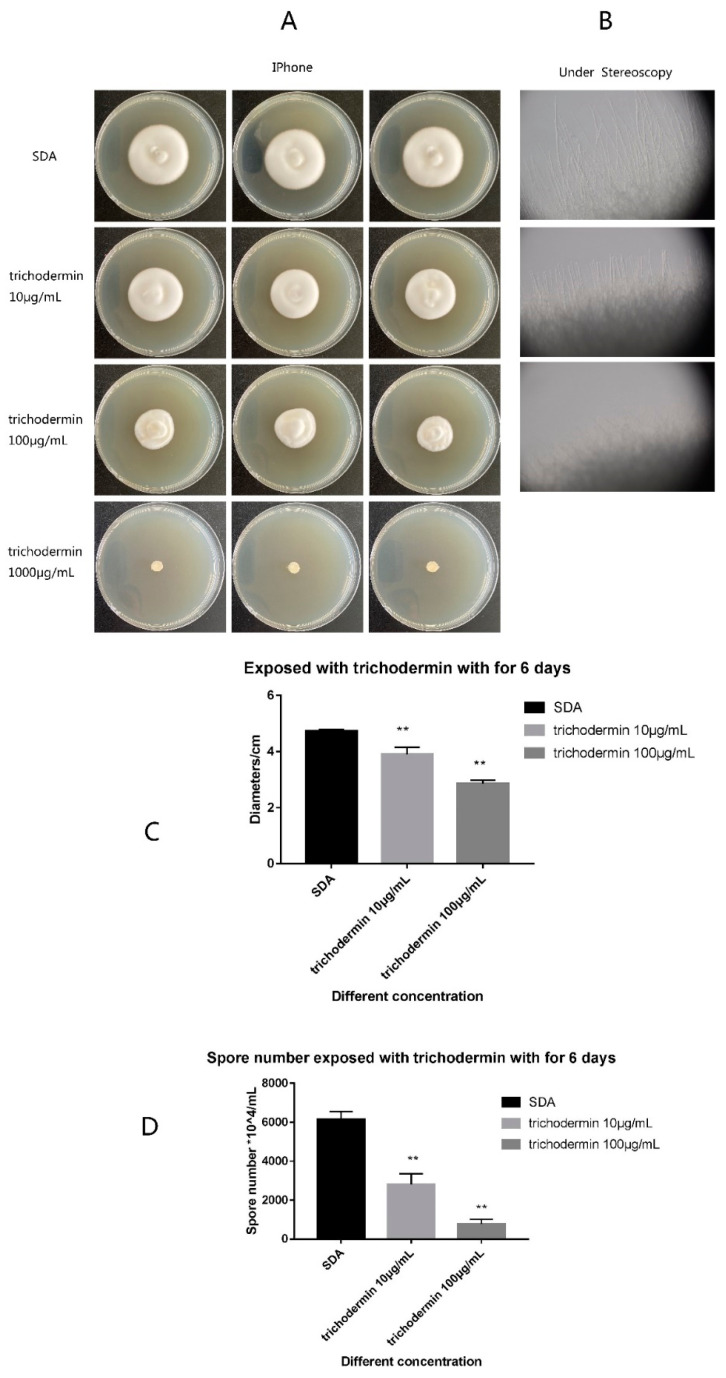
Colony growth and spore germination of *T. mentagrophytes* under trichodermin. After 6 days of culture, the colony sizes and spore numbers of *T. mentagrophytes* were examined. (**A**) 1000 μg/mL trichodermin inhibited the colony growth of *T. mentagrophytes* completely. With decreasing concentration of trichodermin, *T. mentagrophytes* colonies increased but remained significantly smaller compared with the SDA (blank control) group. (**B**) Under a stereoscope, it was found that 10 μg/mL and 100 μg/mL trichodermin inhibited the hyphal growth of *T. mentagrophytes*. The diameters and numbers of *T. mentagrophytes* spores in the 100 μg/mL and 10 μg/mL groups were markedly reduced compared with the control values (*p* < 0.01) (**C**,**D**).** stand for *p* < 0.01.

**Figure 2 jof-08-01006-f002:**
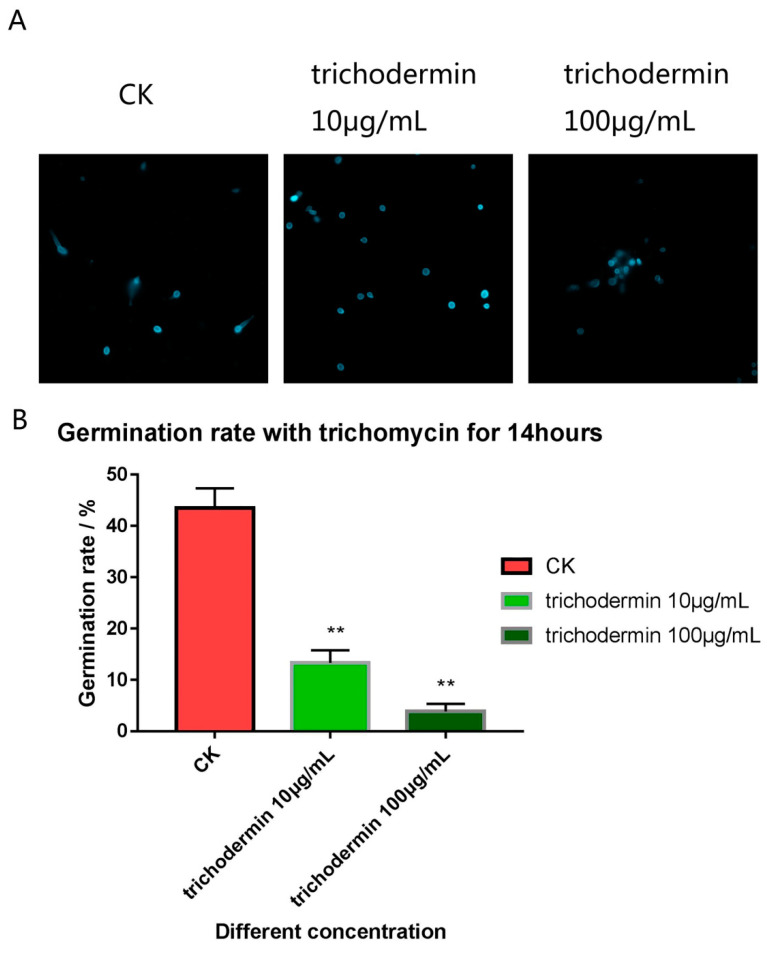
Spore germination observed by fluorescence microscopy after treatment with trichodermin for 14 h and stained with Calcofluor white (**A**). The spore germination rates were correlated with the concentration of trichodermin (**B**). ** stand for *p* < 0.01.

**Figure 3 jof-08-01006-f003:**
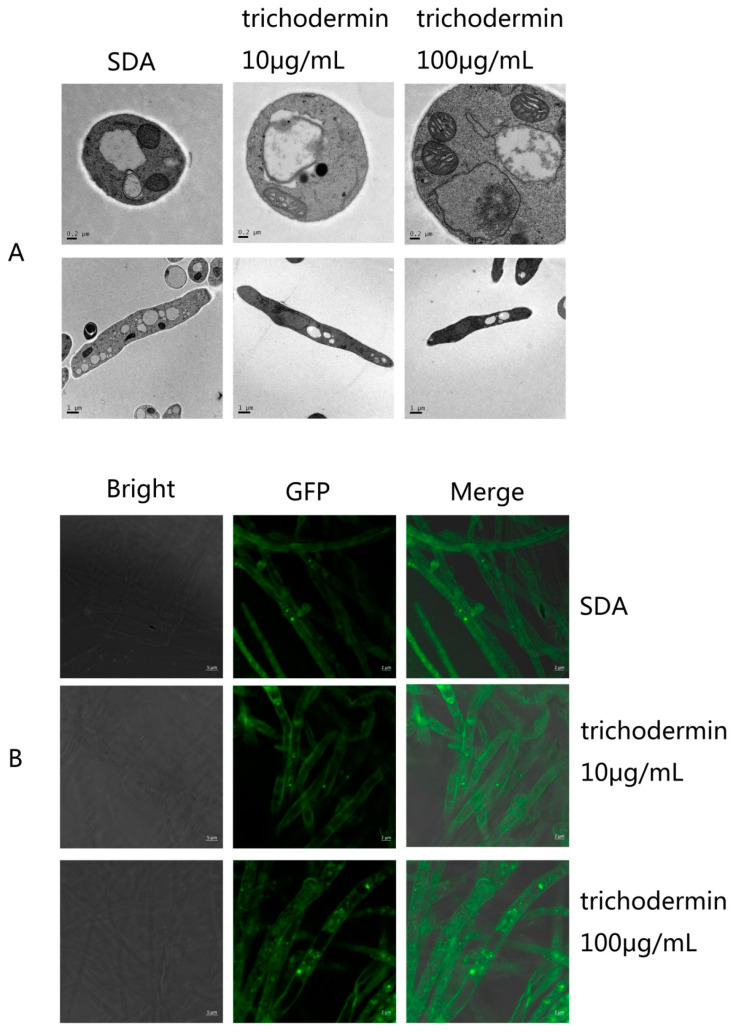
TEM analysis and fluorescent staining of mitochondria in *T. mentagrophytes* treated with trichodermin. (**A**) TEM analysis showing the mitochondria were swollen after treatment with trichodermin. (**B**) Mitochondrial staining with MitoTracker Green^FM^ and detection by confocal microscopy. After treatment with 100 μg/mL trichodermin, the number of mitochondria in *T. mentagrophytes* mycelia increased significantly.

**Figure 4 jof-08-01006-f004:**
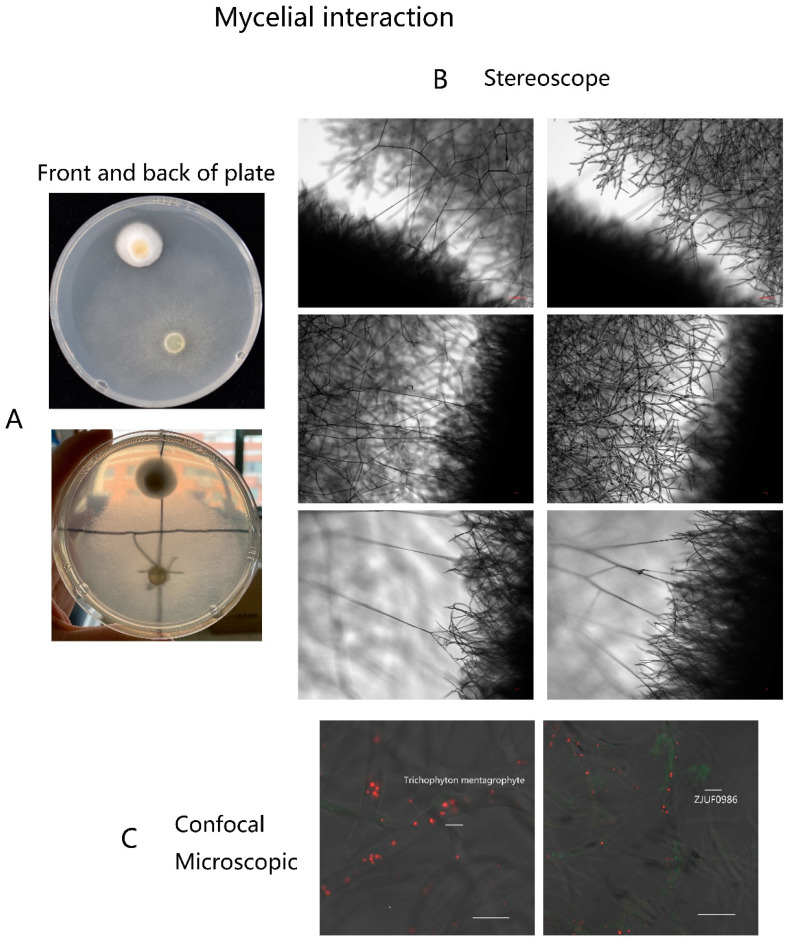
Interaction of *T. taxi* to *T. mentagrophytes*. (**A**) After co-culture for 3 days, the morphology of colony (*T. taxi* and *T. mentagrophytes*) interaction was obtained by photography. (**B**) After co-culture for 3 days, the hyphae of *T. taxi* extended to and intertwined with *T. mentagrophytes* mycelia and invaded the *T. mentagrophytes* hyphae. (**C**) GFP-labeled *T. taxi* and RFP-PTS1-labeled *T. mentagrophytes* were co-cultured and observed under a confocal microscope. The hyphae of both strains were intertwined and interconnected.

**Figure 5 jof-08-01006-f005:**
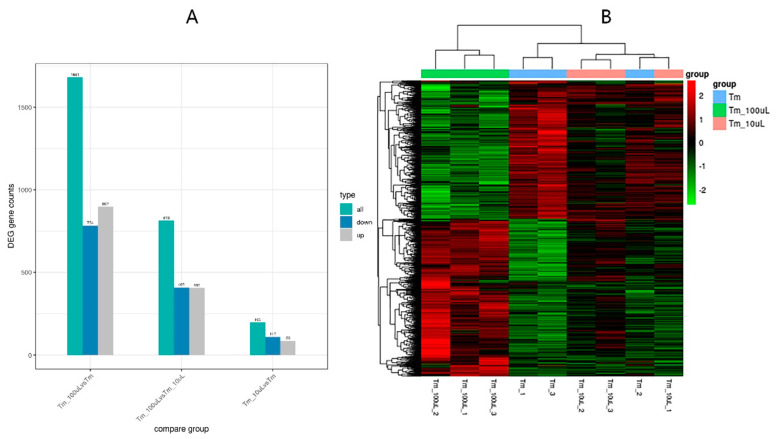
Transcriptome analysis. (**A**) Totally 1681 DEGs were found in the 100 μg/mL trichodermin group compared with the control group, including 897 up-regulated and 784 down-regulated genes (*p* < 0.05); 813 DEGs were found after treatment with trichodermin at 100 μg/mL compared with the 10 μg/mL trichodermin group, including 405 up-regulated and 408 down-regulated genes (*p* < 0.05); 196 DEGs were found between the 10 μg/mL trichodermin and control groups, including 86 up-regulated and 110 down-regulated genes (*p* < 0.05). (**B**) Heat map analysis of the identified genes (*p* < 0.05) is shown based on hierarchical clustering of DEGs in the 100 μg/mL trichodermin, 10 μg/mL trichodermin, and control groups. Gene expression profiles were similar in the 10 μg/mL trichodermin and control groups, while the 100 μg/mL trichodermin group was obviously different from the control and 10 μg/mL trichodermin groups (*p* < 0.05). Red, up-regulation; green, down-regulation.

**Figure 6 jof-08-01006-f006:**
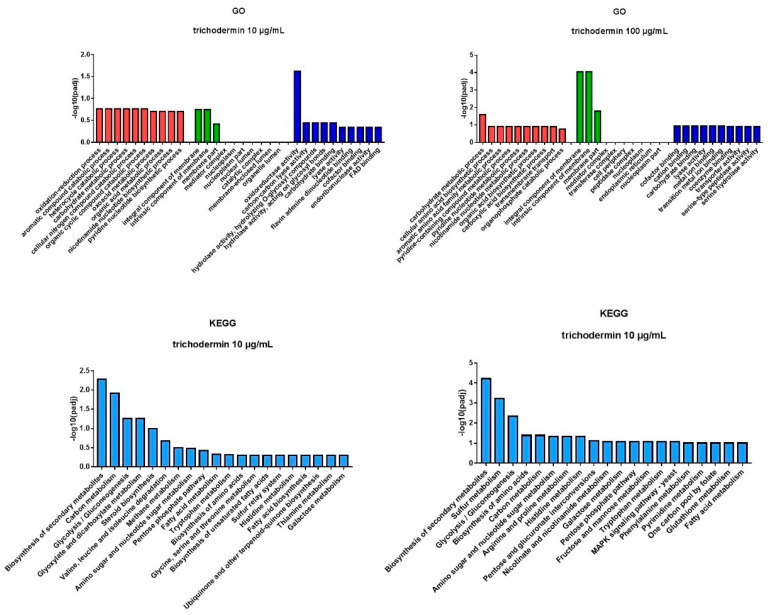
DEG enrichment by gene ontology (GO) terms and KEGG pathways. Based on GO analysis, treatment with trichodermin at 100 μg/mL resulted in enrichment in GO terms such as carbohydrate metabolic process, integral component of membrane, intrinsic component of membrane, and carbohydrate binding and transporter activity. The most enriched GO terms after treatment with trichodermin at 10 μg/mL were carbohydrate metabolic process, integral component of membrane, intrinsic component of membrane, and carbohydrate binding. The most enriched KEGG pathways in the 100 μg/mL and 10 μg/mL trichodermin treatment groups were biosynthesis of secondary metabolites, glycolysis/gluconeogenesis, carbon metabolism, pentose phosphate pathway, and fatty acid metabolism.

**Figure 7 jof-08-01006-f007:**
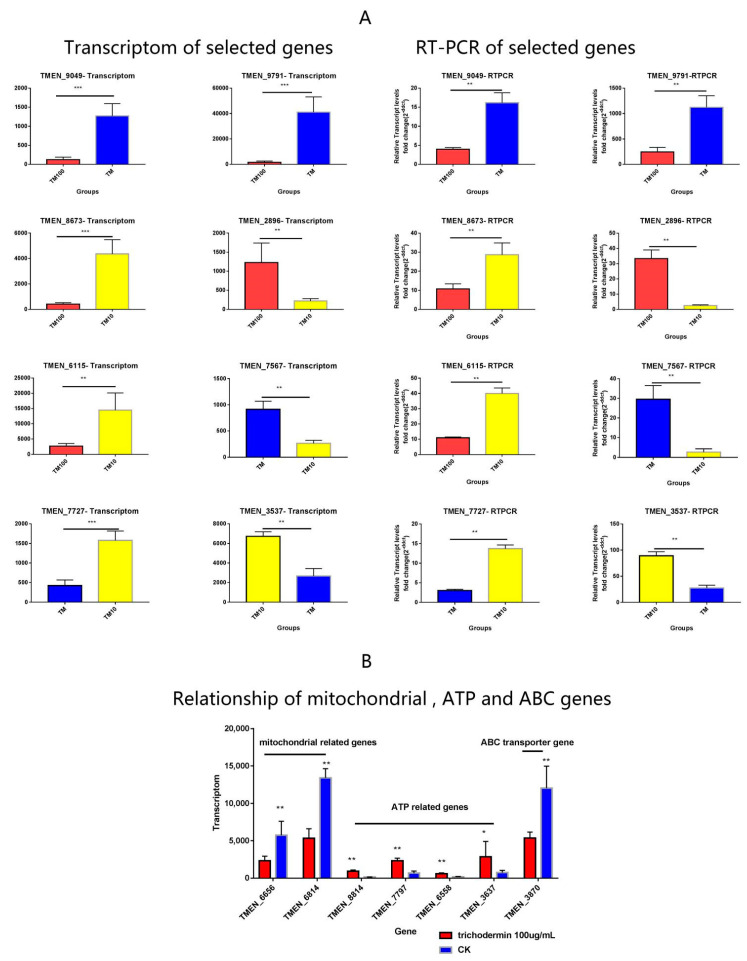
Verification of the transcriptomic data by qRT-PCR. (**A**) The gene expression changes in qRT-PCR corroborated well with transcriptomic data, indicating the transcriptomic data were reliable and could be used for further analysis. (**B**) The analysis demonstrated mitochondria-related genes were significantly up-regulated after treatment with 100 μg/mL trichodermin, while ATP-related genes were down-regulated; the ABC transporter gene TMEN_3870 was down-regulated under trichodermin exposure at 100 μg/mL (*p* < 0.05). *** stand for *p* < 0.001, ** stand for *p* < 0.01; * stand for *p* < 0.05.

**Figure 8 jof-08-01006-f008:**
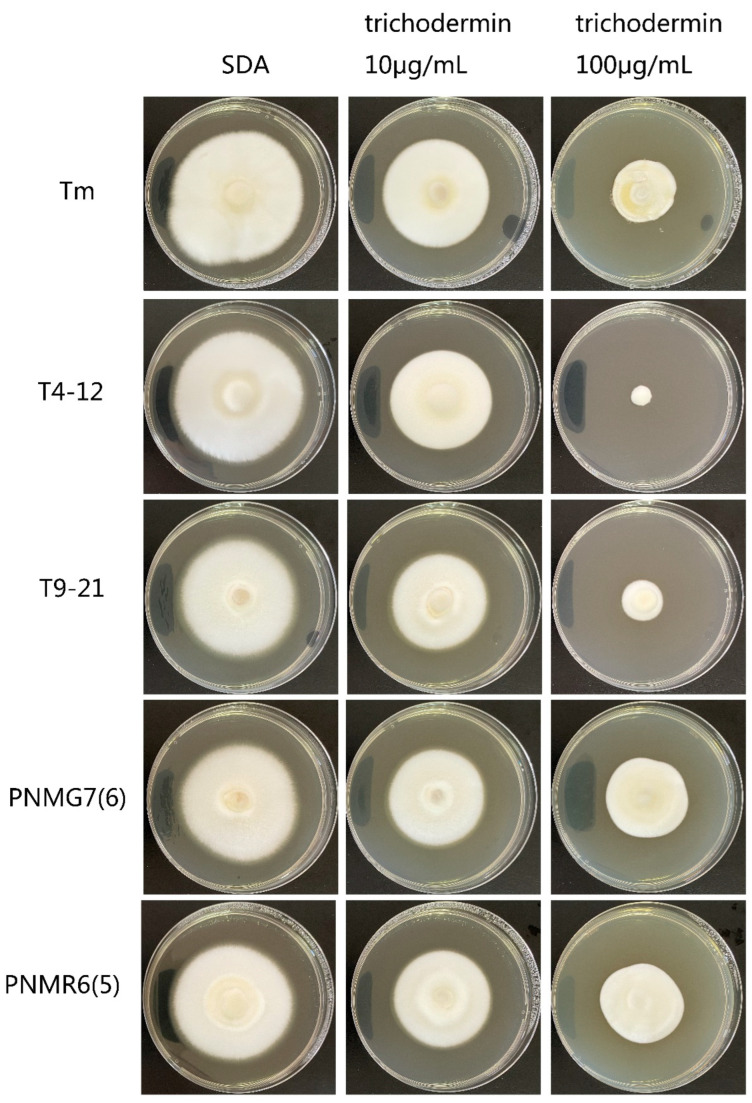
Involvement of the *CDR1* gene (TMEN_6461) in the sensitivity of *T. mentagrophytes* to trichodermin. After 6 days of culture, the CDR1 deletion mutants T9-21 and T4-12 were significantly inhibited by 10 and 100 μg/mL trichodermin compared with wild-type strains. Meanwhile, the hypersensitivity of the mutants was reduced in the *CDR1* complemented strains PNMNR6 (5) and PNMG7 (6).

**Figure 9 jof-08-01006-f009:**
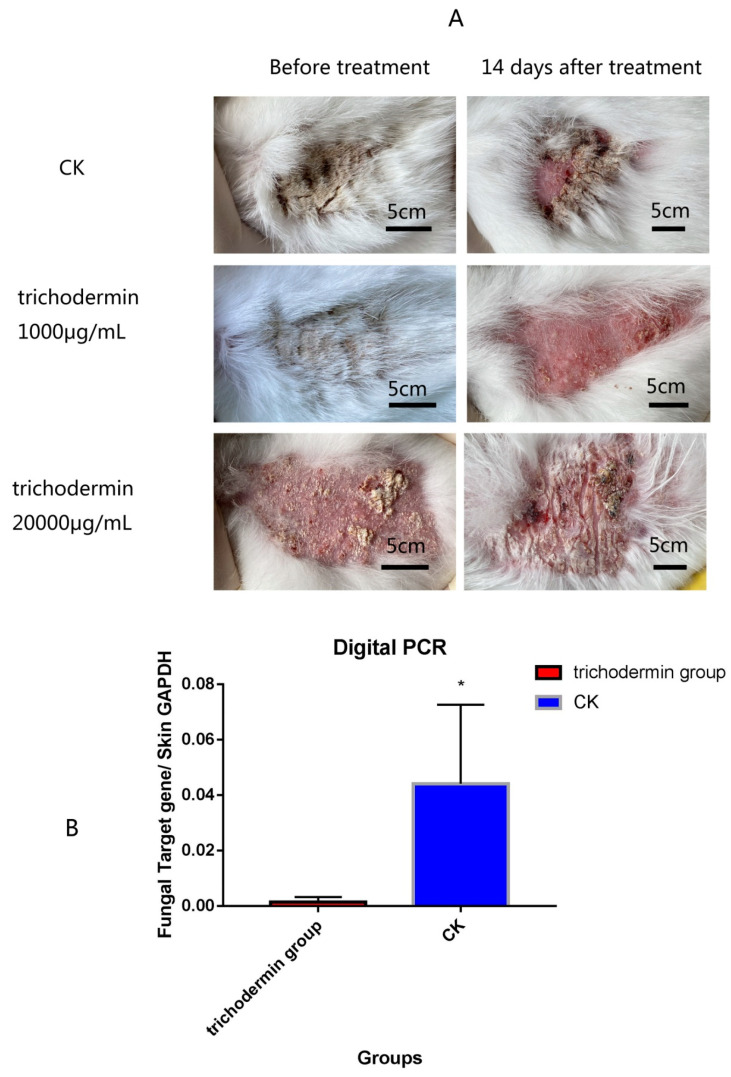
Effects of trichodermin on *T. mentagrophyte* disease on rabbit skin. After treatment by smearing of 1000 µg/mL trichodermin, the wound healing process of *T. mentagrophytes* skin disease was enhanced significantly. However, trichodermin at more than 20 mg/mL was toxic to the skin (**A**). Digital PCR showed that *T. mentagrophytes* in the skin tissue was eliminated by treatment with *trichodermin* (**B**). * stand for *p* < 0.05.

**Table 1 jof-08-01006-t001:** Primers used in qRT-PCR.

Target Genes	Primer	Sequences
TMEN_9049	Forward	GGAGGCCCTTCAGCATTTCT
	Reverse	CCTAAATGGCGACTACCCCC
TMEN_9791	Forward	GTCCGCAGATAAACTGGCCT
	Reverse	TGTCACAAATGGTGGAGGCA
TMEN_8673	Forward	CCCTACACCCAGCTCTGTTG
	Reverse	ACTCGCTGAAAAGAAGGCCA
TMEN_2896	Forward	CAGAAAGGCCCTCAACGAGT
	Reverse	CCCAGAACTTCTTCCCGGTC
TMEN_6115	Forward	CTCCCCTCAGCTGATTCCAC
	Reverse	CAGTCCCTCCTTGTCCATCG
TMEN_7567	Forward	TTTCGTTGTCCAGGCTCCAA
	Reverse	CTTTTGCGCAGCGGAATCTT
TMEN_7727	Forward	CTCACGCCCTTGTCCTGAAT
	Reverse	TGGAAGCCTTTTCCTGCGAT
TMEN_3537	Forward	TTCAAGCTGAGAATCCGGCA
	Reverse	TTCATTGAGCTGGCCAAAGC

**Table 2 jof-08-01006-t002:** Primers and probes used in digital PCR.

Targets	Primer	Sequences
Fungal (Target1)	Forward	CTGCATCCATGGGTTTAGCC
	Reverse	GGATGCTACCTACGGATGCT
	Probe1	FAM-CAGTTTACCTCCCACTGAGTCCGGCA-BHQ1
GAPDH	Forward	TCCTTACCCACTCGTTGTCC
	Reverse	CAACAGGCCAACGTCAAGAA
	Probe2	VIC-AACTCGCAATAGCCGCCAAGGTCCT-BHQ1

**Table 3 jof-08-01006-t003:** Reaction system in digital PCR.

Component	Volume (µL)	Final Concentration
Two-fold digital PCR reaction premix	11	1
Forward primer (10 µM)	1.0	500 nM
Reverse primer (0 µM)	1.0	500 nM
DNA template	2.0 (<10^5^ copies)	
Ribozyme-free water	6.5	
Total volume	22.0	

**Table 4 jof-08-01006-t004:** Inhibition rates of colonial growth, sporulation, and spore germination.

	SDA (CK)	Trichodermin (10 µg/mL)	Trichodermin (100 µg/mL)
Colony diameter (cm)	4.7 ± 0.2	3.9 ± 0.1	2.9 ± 0.2
Colony growth inhibition rate		18.2 ± 2.1%	40.1 ± 3.2%
Number of spores per plate	6.1 ± 0.3 × 10^7^	2.8 ± 0.2 × 10^7^	0.77 ± 0.1 × 10^7^
Sporulation inhibition rate		54.1 ± 2.4%	88.3 ± 3.3%
Spore germination rate	43.5 ± 2.5%	13.3 ± 1.2%	3.9 ± 2.0%
Spore germination inhibition rate		69.5 ± 3.2%	91.2 ± 3.5%

## Data Availability

The transcriptomic data have been submitted to GEO (GEO accession numbers: BioProject, PRJNA785595; SRA, SRP348930). The datasets utilized and/or analyzed in this study are available from the corresponding author upon reasonable request.

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
