# Peer review of "Inhibitory Effect and Mechanism of Trichoderma taxi and Its Metabolite on Trichophyton mentagrophyte"

_jof, 2022, doi:10.3390/jof8101006_

Round 1

Reviewer 1 Report

The publication is very interesting and multi-tracked, but it still needs to be significantly improved.

General remarks:

The authors based the introduction on 2 publications, the second of which is so erroneously cited (m.in the name of the journal) that it is not detectable. Anyway, it is similar in position e.g. 10. Thus, the mere lack of diligence in the preparation of, after all, very short literature, does not make us optimistic about the entire publication.

Definitely, the authors should include a very detailed procedure for the preparation of the trichodermin preparation, especially since due to the animals being tested.

Definitely, since there are microscopic interactions, it would be worthwhile to have a plate test evaluating the direct interaction in biotic tests.

Figure 6 is completely unreadable.

Giving in the discussion generally known information about the abilities of Trichoderma, as a factor of biocontrol, without any literature reference, especially from the last 2-3 years, indicating the latest data, is somewhat strange.

The work requires careful stylistic and editorial improvement.

Author Response

Comments and Suggestions for Authors

The publication is very interesting and multi-tracked, but it still needs to be significantly improved.

General remarks:

The authors based the introduction on 2 publications, the second of which is so erroneously cited (m.in the name of the journal) that it is not detectable. Anyway, it is similar in position e.g. 10. Thus, the mere lack of diligence in the preparation of, after all, very short literature, does not make us optimistic about the entire publication.

Author response: We are sorry for this weak point, we have corrected it and added several related publications. Please check.

  1. Li HY, Wang L, Zhao ZW: Studies on the endophytic fungi and antifungal activity of Jatropha curcas L Research and development of natural products 2006, 18(1):78-80.
  2. Harman GE, Howell CR, Viterbo A, Chet I, Lorito M: Trichoderma species—opportunistic, avirulent plant symbionts. Nature reviews microbiology 2004, 2(1):43-56.
  3. Kubicek CP, Harman GE: Enzymes, Biological Control, and Commercial Applications: Taylor & Francis; 1998.

Definitely, the authors should include a very detailed procedure for the preparation of the trichodermin preparation, especially since due to the animals being tested.

Author response: A very detailed procedure for the preparation of the trichodermin preparation and related reference have been listed below. Please check.

Author response: the preparation process of trichodermin is as follows:

After the strain ZJUF0986 is activated on PDA medium, the fungal cake at the edge of the colony is cut with a punch, and then it is connected to the liquid seed medium for culture and fermentation.

The crude petroleum ether extract of the fermentation broth is dissolved in petroleum ether, and then back extracted with deionized water. Anhydrous sodium sulfate is added to the organic phase and dried. After filtration, it is concentrated under reduced pressure at 50 ° C to obtain a colorless viscous substance. Dissolve the colorless viscous substance with chloroform, filter it, mix it with H60 thin layer chromatography silica gel, volatilize the solvent, install the column, and elute with chloroform / methanol mobile phase with different gradient. The collected eluates of each unit were concentrated and prepared into methanol solution. The spots were tracked by high-performance thin layer chromatography (TLC). The samples with the same spots and antibacterial activity were combined to separate and purify the active metabolites.

Related Reference:

Wang G, Lu S, Zheng B, Zhang C, Lin F: Control of rice sheath blight with the endophytic fungus ZJUF0986 and its bioactive metabolite. Chinese Journal of Biological Control 2009, 25(1):30-34.

Definitely, since there are microscopic interactions, it would be worthwhile to have a plate test evaluating the direct interaction in biotic tests.

Author response: We thank Reviewer for this suggestions, we have added the photos. Please check in Figure 4.

Figure 6 is completely unreadable.

Author response: We have revised Figure 6 Please check. Thank you very much.

Giving in the discussion generally known information about the abilities of Trichoderma, as a factor of biocontrol, without any literature reference, especially from the last 2-3 years, indicating the latest data, is somewhat strange.

Author response: We are sorry for this weak point, we have corrected it and added several related publications. Please check.

The work requires careful stylistic and editorial improvement.

Author response: We are sorry for this weak point, we have revised the manuscript. Please check.

Reviewer 2 Report

The MS presents activity of trichodermin on T. mentagrophytes and describes potential mechanisms along with in vivo experiments.

Although authors present a lot of data some of them are not easy to follow.

Major concern:

1. How did authors selected amount to be used in first experiments? And why you ommited the largest dose in results?

2. How you prepared trichodermin, or did you used commercially available chemical?

3. Regarding in vivo experiment - why you used 1000 ug/ml and 20 mg/mL; why only treated once a day (as you know most of the treatments is used several times a day).

4. How you maintained the animals? Do you have ethical approval?

5. When you ceased the experiment in case of toxic effect?

Author Response

Comments and Suggestions for Authors

The MS presents activity of trichodermin on T. mentagrophytes and describes potential mechanisms along with in vivo experiments.

Although authors present a lot of data some of them are not easy to follow.

Major concern:

  1. How did authors selected amount to be used in first experiments? And why you omitted the largest dose in results?

Author response:

This is because we have done a preliminary experiment, and a smaller concentration of Trichoderma cannot play a significant inhibitory role on Trichophyton. And we use several doses in first experiment. At 1000μg of trichodermin, fungi were completely inhibited, so there was no data on the maximum concentration. Only have 10μg of trichodermin and 100μg of trichodermin

  1. How you prepared trichodermin, or did you used commercially available chemical?

Author response:

After the strain ZJUF0986 is activated on PDA medium, the fungal cake at the edge of the colony is cut with a punch, and then it is connected to the liquid seed medium for culture and fermentation.

The crude petroleum ether extract of the fermentation broth is dissolved in petroleum ether, and then back extracted with deionized water. Anhydrous sodium sulfate is added to the organic phase and dried. After filtration, it is concentrated under reduced pressure at 50 ° C to obtain a colorless viscous substance for standby. Dissolve the colorless viscous substance with chloroform, filter it, mix it with H60 thin layer chromatography silica gel, volatilize the solvent, install the column, and elute with chloroform / methanol mobile phase with different gradient. The collected eluates of each unit were concentrated and prepared into methanol solution. The spots were tracked by high-performance thin layer chromatography (TLC). The samples with the same spots and antibacterial activity were combined to separate and purify the active metabolites.

We do not use commercial trichodermin.

Related Reference:

Wang G, Lu S, Zheng B, Zhang C, Lin F: Control of rice sheath blight with the endophytic fungus ZJUF0986 and its bioactive metabolite. Chinese Journal of Biological Control 2009, 25(1):30-34.

  1. Regarding in vivo experiment - why you used 1000 ug/ml and 20 mg/mL; why only treated once a day (as you know most of the treatments is used several times a day).

Author response:

We thanks for this point. In animal experiments, the dose of 1000ug is the maximum dose in vitro experiments, and the concentration of 1000ug can reflect the best antifungal effect in vitro. Since we consider the animal experiment operability, the dose of 20mg was chosen. Because there may be waste in the operation, the dosage should be larger

The purpose of treating once a day is to promote the application in the future, and the labor cost of treating once will be the lowest.

  1. How you maintained the animals? Do you have ethical approval?

Author response: Male New Zealand rabbits (5-week-old) weighting 1-1.5 kg were purchased from Zhejiang Animal center belonged to Zhejiang Academy of Agricultural Sciences and acclimatized for 1 week prior to use. And raised in a clean level feeding environment.

This study was approved by the Ethics Committee of the Zhejiang Academy of Agricultural Sciences (ethics protocol no. 002762) and performed in accordance with the principles and guidelines of the Zhejiang Farm Animal Welfare Council of China.

  1. When you ceased the experiment in case of toxic effect?

Author response:

In case of toxic effect, we apply trichodermin once a day for three days.  And then we ceased the application. After the end of the experiment (14 days after treatment with medicine) all the animals were preserved on one side of the cage and euthanized by intravenous injection of 100mg/kg of sodium pentobarbital (Sigma-Aldrich cas no: 57-33-0), resulting in painless death of experimental animals.

Please check in "10. Digital PCR for assessing amount of fungal material in skin tissue".

Reviewer 3 Report

It is a novel idea to use the metabolite of one fungus to inhibit another, since it is true that on numerous occasions they compete for dominance in the infected territory.

It is a well-structured study with numerous in vitro tests that provide a lot of information at the molecular level and a rabbit test that provides a lot of information at the clinical level.

However, there are two issues that must be resolved before accepting the manuscript for publication.

In the first place, there seems to be a lack of toxicity tests (controls) in the work, since we only know that at a clinical level, the highest concentration seems to cause a worse lesion. The authors need to clarify this matter better.

It would also be convenient to properly justify why the in vitro tests are done only at two concentrations (10 ug/ml and 100 ug/ml) and the in vivo tests at 1000 ug/ml and 2000 ug/ml.

Author Response

Comments and Suggestions for Authors

It is a novel idea to use the metabolite of one fungus to inhibit another, since it is true that on numerous occasions they compete for dominance in the infected territory.

It is a well-structured study with numerous in vitro tests that provide a lot of information at the molecular level and a rabbit test that provides a lot of information at the clinical level.

However, there are two issues that must be resolved before accepting the manuscript for publication.

In the first place, there seems to be a lack of toxicity tests (controls) in the work, since we only know that at a clinical level, the highest concentration seems to cause a worse lesion. The authors need to clarify this matter better.

Author response: After treatment, the skin cracked, which indicates that Trichoderma is still toxic in the case of skin rupture at high concentration.

According to the requirements of the reviewer, we conducted a supplementary blank control test, shaved off the hair on the skin of healthy rabbits, and then applied 1ml of 20mg / mL trichodermin. After three days of smearing, we take photos and observing, we found that 1mL of 20mg / mL trichodermin did not have any effect on the skin of rabbits. It indicates that this dose should be safe and reliable for normal rabbit skin. Please check.

It would also be convenient to properly justify why the in vitro tests are done only at two concentrations (10 μg/mL and 100 μg/mL) and the in vivo tests at 1000 μg/mL and 20000μg/mL.

Author response: We thanks for this point. The in vitro tests are done only at two concentrations (10 μg/mL and 100 μg/mL) becausethey were selected since we did a pre experiment. The concentration in this range can well reflect the antifungal effect of the drug on Trichophyton mentagrophytes, showing a dose-dependent

In animal experiments, the dose of 1000μg is the maximum dose in vitro experiments, and the concentration of 1000μg can reflect the best antifungal effect in vitro. Since we consider the animal experiment operability, the dose of 20mg was chosen. Because there may be waste in the operation, the dosage should be larger.

Round 2

Reviewer 1 Report

The revisions have improved the publication and I wish the authors success in their continued research.

Author Response

We thanks the reviewer for the positive comments.

Reviewer 2 Report

.

Author Response

(The authors gave the same response as above.)
